# The Immune System of Marine Organisms as Source for Drugs against Infectious Diseases

**DOI:** 10.3390/md20060363

**Published:** 2022-05-28

**Authors:** Alberto Falco, Mikolaj Adamek, Patricia Pereiro, David Hoole, José Antonio Encinar, Beatriz Novoa, Ricardo Mallavia

**Affiliations:** 1Institute of Research, Development, and Innovation in Healthcare Biotechnology in Elche (IDiBE), Miguel Hernández University (UMH), 03202 Elche, Spain; jant.encinar@umh.es (J.A.E.); r.mallavia@umh.es (R.M.); 2Fish Disease Research Unit, Institute for Parasitology, University of Veterinary Medicine, 30559 Hannover, Germany; 3Institute of Marine Research, Consejo Superior de Investigaciones Científicas (IIM-CSIC), 36208 Vigo, Spain; patriciapereiro@iim.csic.es (P.P.); beatriznovoa@iim.csic.es (B.N.); 4School of Life Sciences, Keele University, Keele ST5 5BG, UK; d.hoole@keele.ac.uk

**Keywords:** innate immune system, antimicrobial drug discovery, antimicrobial peptides, antimicrobial metabolites

## Abstract

The high proliferation of microorganisms in aquatic environments has allowed their coevolution for billions of years with other living beings that also inhabit these niches. Among the different existing types of interaction, the eternal competition for supremacy between the susceptible species and their pathogens has selected, as part of the effector division of the immune system of the former ones, a vast and varied arsenal of efficient antimicrobial molecules, which is highly amplified by the broad biodiversity radiated, above any others, at the marine habitats. At present, the great recent scientific and technological advances already allow the massive discovery and exploitation of these defense compounds for therapeutic purposes against infectious diseases of our interest. Among them, antimicrobial peptides and antimicrobial metabolites stand out because of the wide dimensions of their structural diversities, mechanisms of action, and target pathogen ranges. This revision work contextualizes the research in this field and serves as a presentation and scope identification of the Special Issue from *Marine Drugs* journal “The Immune System of Marine Organisms as Source for Drugs against Infectious Diseases”.

## 1. The Clinical Arsenal of Antimicrobial Agents Requires an Update

The continuous emergence of bacterial resistances to antibiotics and deadly viral outbreaks that may result in severe pandemics are seriously damaging current health standards worldwide.

The threat of multidrug resistance (MDR) bacteria already exerts a significant impact on citizens’ welfare, clinical resources, and associated costs [1,2]. MDR bacteria are those bacterial strains that are not susceptible to one or more antimicrobials from three or more different antimicrobial classes, such as extensively drug-resistant (XDR) and pandrug-resistant (PDR) [3]. The main representatives of MDR bacterial pathogens, apart from the causative agent of tuberculosis *Mycobacterium tuberculosis*, are encompassed in a group so-called ESKAPE, i.e., *Enterococcus faecium*, *Staphylococcus aureus*, *Klebsiella pneumoniae*, *Acinetobacter baumanii*, *Pseudomonas aeruginosa*, and *Enterobacteriaceae* [2]. As a response to this issue, numerous health agencies and societies, such as the Infectious Diseases Society of America (IDSA) and the World Health Organization (WHO), have drawn attention to the lack of new antibiotics and the urgent need for cost-effective treatments for decades [4,5,6].

Despite these facts, only a few new classes of antibiotics have been discovered in the past 60 years, which contrasts with the over 20 classes identified and marketed between 1930 and 1962. This dramatic reduction in the discovery of novel classes of antibiotics (i.e., with a completely novel mechanism of action) since 1962, when diaminopyrimidines (dihydrofolate reductase inhibitors) were introduced, is referred to as the ‘innovation gap’. This gap has been only partly alleviated recently with clinical approval being granted for the use of the following new antibiotic classes in humans: oxazolidinones (Linezolid as the first approved agent in the class, 2000), lipopeptides (Daptomycin, 2003), pleuromutilins (Retapamulin, 2007), tiacumicins (Fidaxomicin, 2011), and diarylquinolines (Bedaquiline, 2012) [5,7,8]. Unfortunately, the majority of the efforts of pharmaceutical companies have focused on the development of analogs because no one expected such a rapid bacteria counterattack to reduce most of the antibiotic analogs’ efficiencies so easily, putting the antibiotic innovation rate far behind the rate of emergence of bacterial resistances. Consequently, most classes of antibiotics have substantial resistance problems today, raising fears for the return of a pre-antibiotic-like scenario [1,4,5,7]. Among the reasons contributing to this long-lasting drought in the discovery of new antibiotics, it is worth mentioning (i) their low-return expectations, which discourage pharmaceutical companies from investing in their research and development, and (ii) the usually trend-driven public funding schemes concentrating on immediate issues, independently of the long-term impact [4,6,7].

The general picture becomes even bleaker when also considering the incidence of viral diseases. These are particularly troublesome because of (i) the shortage of effective antivirals against most of them (including common ones already globally spread), (ii) the development of resistances to existing antivirals, and (iii) the growing number of emerging pathogenic viruses or strains [9,10,11,12]. Certainly, this is due to the fact that since the discovery of the first vaccine at the end of the XVIII century, progress has almost entirely relied upon the development of vaccines to tackle viral diseases. However, the recent unfortunate events with the severe acute respiratory syndrome coronavirus 2 (SARS-CoV-2) pandemic showed the world’s limited capacity for rapid action against an unknown virus.

As a result of this strategy, together with the poor support provided by the pharmaceutical sector and public funding (as it occurs for the antibacterial drugs), it is not surprising that the current arsenal of antivirals is so limited that the WHO has repeatedly demanded urgent attention in this regard. Indeed, there are only 118 antiviral drugs approved by the US Food and Drug Administration (FDA) until 2020, out of which 52 are aimed at human immunodeficiency virus 1 (HIV1); 19 and 8 at hepatitis C (HCV) and B (HBV) viruses, respectively; 11 at herpes simplex virus (HSV); 9 at influenza virus; 6 at human cytomegalovirus (HCMV); 5 at human papillomavirus (HPV); 5 at varicella-zoster virus (VZV); and 3 at respiratory syncytial virus (RSV) [13]. Among them, molnupiravir and remdesivir were recently repurposed for also treating coronavirus disease 2019 (COVID-19), caused by SARS-CoV-2 [14]. Present therapies are therefore scarce, generally expensive, and require an early diagnosis because they are either too specific and/or only effective if delivered at the right time, often during a short lapse of the disease progression. Some other antivirals are for chronic illnesses and then prescribed long term or even for life, which generally implies less patient compliance and consequently lower efficiency rates [15].

## 2. Current General Strategies for the Discovery of Antimicrobials

Although the described above scenario is disappointing, there is confidence that new antimicrobial drugs will be available to support medicine in safeguarding and improving health standards for the future. As representative examples of such optimism, there are the aforementioned discoveries of some new classes of antibiotics in the past 20 years [5,7,8] and the development of very effective antiviral treatments for HIV1 and HCV [11,12,15]. Such progress can be largely explained by the increasing sophistication of the new strategies for drug discovery, particularly with regard to the two main general aspects of the molecule screening procedure, i.e., the methodology and the source of compounds, which are often interdependent [6,16,17].

Regarding the methodology employed, despite the latter developments of automated massive screening systems for in vitro antimicrobial trials, it is still not feasible in this way to cope with the dimensions of testing the huge molecular libraries against all the present and continuously appearing and evolving list of pathogens. Thus, this task is being complemented by hypothesis-driven strategies targeting precisely identified biologically relevant and clinically significant sites [6,17]. Among these new strategies, those involving in silico approaches have greatly grown in popularity and expectancies, as they allow the guided search of such compounds by progressively more powerful computational methods [17]. In general terms, they consist of the determination of the theoretical binding affinity of large amounts of compounds (from which structural data are compiled in extensive chemical libraries) to functionally relevant regions from critical proteins involved in essential molecular pathways for the living of the target microorganisms. For instance, in this way, molecular docking and dynamics simulations targeting catalytic and regulatory allosteric sites for searching potential competitive and noncompetitive inhibitors, respectively, have been carried out in our groups [18,19]. Likewise, there is the alternative possibility of designing peptides equivalent to relevant lineal interacting-protein regions (i.e., not conformationally arranged), with the final goal of competing for the binding sites from the protein counterparts and thus disrupting the particular molecular pathway [20,21]. For both approaches, detailed molecular structure data of the target isolated proteins and their complexes are required, as well as knowledge of the molecular pathways of both the cellular response and the pathogen infection involved in each particular case [18,19,20,21].

Although advances in the methodology used have shown great utility in optimizing material resources and reducing execution and analysis times to cover extensive screenings, for significantly increasing the total throughput (i.e., enlarging the chemical space), it is also crucial to make a promising initial selection of the source of molecules to be analyzed. In order to choose molecular populations with high possibilities of containing successful compounds for the proposed objective, scientists searched popular knowledge for successful proven ancestral remedies to identify and characterize the active moieties. Consequently, approximately half of the commercially available drugs today are natural products (mostly from plants) and their semisynthetic derivatives, as their chemical structures are commonly susceptible to analog development [16,22].

In this sense, advances in basic research have also largely contributed to broadening and more precisely fractionating the range of supplying sources of molecules to be tested, as well as replicating their complex structures by chemical synthesis [16,23]. However, the initial discovery of most of these compounds has been conditioned by the accessibility to their sources. As a consequence, most exploration has been carried out in terrestrial habitats, but scarcely in the most biodiverse marine environments. In this regard, the improvements in marine prospection technologies are changing this trend [17,24].

## 3. Marine Habitats as Promising Sources of New Antimicrobials

Marine ecosystems open a wide horizon with great potential for the discovery of new antimicrobials of biological origin, primarily because they are the most extensive and environmentally diverse habitats colonized by living beings on Earth. In addition, since they are considered to be the oldest environments from which life originated and expanded, they currently harbor the greatest biodiversity, as well as biomass [25]. In tune with this fact, it has been suggested that the higher biodiversity, the broader the chemical variety within such ecosystems [26]. Consequently, more than two-thirds of the known molecular scaffolds have been found unique to marine organisms, highlighting their great potential as a source of therapeutic agents [27,28]. In addition, their obtention, when synthetic chemistry procedures were unfeasible, could be conducted through cell culture and aquaculture production systems, with the eventual support of marine biotechnology, in order to release exploitation pressure from wild ecosystems.

Accordingly, microorganisms, including pathogenic ones, are more abundant in marine environments [29,30]. However, despite being continuously exposed to these microbial-rich challenging environments, marine organisms keep healthy under normal circumstances through a complex network of defense mechanisms that have had to co-evolve under such conditions of selective pressure. These protective mechanisms have traditionally been classified into two main systems, so-called innate and adaptive immune systems, which are interconnected, coordinated, and interdependent to make such artificial differentiation only useful for didactic purposes. Among the constituent fractions of these systems, the humoral aspect within the innate immunity is of special interest, as not only is it present in all existing organisms, but it also contains a wide variety of molecules endowed with wide antimicrobial activity and thus has the potential for being successfully implemented in biomedical applications targeting human and animal infectious diseases [31,32,33].

Such activity can be exerted by directly targeting and inactivating the pathogen and/or by indirectly acting on the host’s cells in such a way as to confer protection against a particular infection. Since such direct activity is usually targeted at molecular patterns common to many families of microorganisms, these molecules have high potential for exploitation to combat pathogens affecting other animal groups. On the other hand, the indirect (immunomodulatory) effect could be more limited in this sense, although it should not be totally ruled out since many of the target molecular pathways and their participants are highly conserved evolutionarily and still keep relatively unaltered the sites modulated by these compounds. It is also worth mentioning that since these molecules do not act on the pathogen, and the induced immune response is multitarget, not only would they not contribute to the selection of resistant mutants, but they would also exert a broad-spectrum antimicrobial effect [34,35,36,37].

In this vein, the high abundance of these immune molecules in aquatic environments is not a coincidence, as the vast majority of organisms are invertebrates in which the innate immune system predominates [31,38]. In addition, fish are by far the largest and more diverse class of vertebrates [39], and, although they have one of the earliest forms of adaptive immunity, they still largely rely on their innate immune responses [40,41]. Their complex mucosal barrier systems are relevant representatives supporting this argument [40,41,42], as they are one of the major producers of antimicrobial peptides (AMPs) [41,43].

Furthermore, the activity of the immune system, whether it comes from organisms with one or both arms of the immune response, is highly modulated by other intervening molecules from, mostly, hormonal and metabolic pathways. An increasing number of scientific contributions are drawing attention to the influence of metabolism on the immune system and how many primary and secondary metabolites, either present in the host or the host’s microbiome [36] or other independent organisms [44], retain antimicrobial properties capable of boosting the immune response and thus acting as indirect effectors [35].

## 4. Antimicrobial Peptides (AMPs)

AMPs can work as a representative example of immune molecules that are widely distributed in aquatic habitats and have the potential to be applied to a clinical setting. Also known as host defense peptides (HDPs) [45], these gene-coded multitarget molecules play the role of effectors of the innate immune system in all organisms, and hence, they have been shaped by the selective pressure of host–pathogen interactions to comprise multimodal direct activities against bacteria, fungi, and/or enveloped and nonenveloped viruses. Some of them are additionally endowed with potent immunomodulatory properties, which altogether contribute to making difficult the development of resistant pathogens [45,46,47].

In general, the cationic and amphipathic nature of AMPs allows them to interact with the commonly negatively charged lipid cell membranes of pathogens, as well as envelopes of viruses [47]. The mechanisms proposed to explain their mode of action include different pore-formation models to destabilize/permeabilize the pathogen membranes [48,49]. Because of this ability to interact with membranes, AMPs were suggested to be included within a broader group called membrane-active peptides (MAPs) together with the family of the so-called cell-penetrating peptides (CPPs) and peptide toxins [39]. In particular, CPPs are membrane-interacting peptides with the ability to translocate into the cell, and many AMPs have been shown to act as CPPs and vice versa [50]. In fact, further to these pore-forming mechanisms, there is evidence of their ability to diffuse towards intracellular targets and act in other ways, such as inhibiting the synthesis of pathogen cell walls, nucleic acids, or proteins, and even reducing pathogen-induced enzymatic activity [51].

Lots of radically different (both linear and disulfide cross-linked) AMPs have been already identified within a wide range of marine organisms, where several unique classes of AMPs have been located such as piscidins in fish, penaeidins, and crustins in crustaceans, tachyplesins in chelicerates, arenicins and nicomicins in cnidarians, and mytilins and myticins in mussels [31,33,52,53,54,55,56,57]. As can be observed in Figure 1, these peptides are frequently amphipathic, and their secondary structures correspond to helical and/or beta structures commonly stabilized by one or more intramolecular disulfide bonds. Among them, the α-helix structure of piscidins stands out due to its highly defined distribution of amphipathicity. Such an arrangement is highly conserved in the piscidin class, in contrast to the limited conservation of the primary sequences of its members, which have been identified as isoform clusters in a large number of fish families throughout the *Actinopterygii* superorder [58,59,60]. Thus, piscidins also encompass chrysophsins, dicentracins, epinecidins, gaduscidins, misgurins, moronecidins, myxinidins, and pleurocidins, whose names refer to the genus they were initially described from, prior to their regrouping within this major class [59,60]. The expression profile of piscidins varies depending on the isoform and species studied. In general, the piscidin pool within an organism is widely distributed, but interestingly, for harvesting purposes, the mucosal tissues such as gills, gut, and skin usually prevail as the main producing sites, and their expression is highly inducible by a variety of stimuli [58,61,62,63,64,65].

In vitro and in vivo studies on marine AMPs have reported potent, broad-spectrum, antimicrobial activity against animal, but also human, pathogens, comprising both Gram-negative and -positive and MDR bacteria [66,67,68,69], viruses [53,60,70,71,72,73], fungi [69,74], and protozoa [69,75]. Moreover, there have been reported synergic effects between fish AMPs, activity when provided orally, and broad interspecies immunomodulatory properties [46,53,66,70,76]. Unfortunately, the high cost of peptide production has restricted the clinical implementation prospects of some AMPs so far [45,77,78,79], but their purification from these natural AMP-rich farmable sources may help overcome this obstacle.

## 5. Antimicrobial Metabolites

Often regarded as the immediate byproduct of a metabolic process, metabolites broadly comprise intermediate or end products of metabolism with either essential (primary metabolites) or accessory (secondary metabolites) functional roles. Thus, antimicrobial metabolites are generally secondary metabolites that protect from infectious diseases by usually modulating the immune defenses, i.e., by more commonly exerting an indirect antimicrobial effect rather than directly on the pathogens. Such an effect has been reported to cross interspecific barriers, often even phylogenetically distant ones. Thus, the discovery/obtention of sources of antimicrobial metabolites can potentially occur in, and apply to, any organism [44,80].

Recently, many studies in the field have focused on commensal microorganisms, mainly gut flora, and their corresponding hosts in order to elucidate the microbiome–host mutualistic cross-talk and also reveal flora’s antimicrobial metabolites that immunomodulate the host and/or exert direct antimicrobial activity against particular pathogens [81]. However, for many decades, extensive research was already performed on the profiling of antimicrobial metabolites produced by marine organisms, mostly microorganisms, algae, and invertebrates [44,80]. Lately, much progress is being made in those from fish and, therefore, with great potential for application in other vertebrates. For instance, the oxysterol 25-hydroxycholesterol (Figure 2a) and its antiviral activity have been described in fish [82,83,84], and although this molecule was already known in higher vertebrates, this fact certainly holds out hope for the identification of new compounds in this group of animals.

Current knowledge has revealed a vast chemical diversity of these bioactive substances in marine organisms, comprising, among others, polyethers, polyols, polyphenols, polysaccharides, saponins, steroids, terpenoids, and unsaturated fatty acids, but also unusual or noncanonical amino acids (i.e., nonproteinogenic amino acids), which can combine together, and with other usual amino acids and nonamino acid moieties, to form bioactive peptides [17,54,55]. Focusing on these peptides in order to differentiate from the above-described AMPs, they are not genetically coded, but directly produced through enzymatic reactions, such as the unusual amino acids [85]. Marine ecosystems have been found rich in these nonribosomally biosynthesized peptides, some of which are endowed with antimicrobial properties, often as of yet unknown mechanism. They are mainly lipopeptides and dipeptides, both containing species with linear and cyclic arrangements, i.e., taking ring structures by stably connecting their end and/or side-chain groups, as well as other cyclic peptides not belonging to these major chemical classes [54,55,57]. The main marine antimicrobial metabolites mentioned in this work are listed in Table 1.

In particular, lipopeptides consist of a peptide linked to a fatty-acid-derived portion. The approved daptomycin and polymyxin antibiotics, initially isolated from terrestrial bacteria species, belong to this class of compounds. Apparently, lipopeptides are predominant compounds (about 40%) in cyanobacteria [111], as well as major ones in other bacteria groups, mainly *Actinobacteria* and *Firmicutes* phyla [112]. As examples of marine antimicrobial lipopeptides, there are tauramamide (Figure 2c) [87] and aneurinifactin [86], discovered in *Brevibacillus laterosporus* and *Aneurinibacillus aneurinilyticus*, respectively, and both reported as antibacterials (Table 1). The mode of action proposed to explain this activity in lipopeptides in general [112], and aneurinifactin in particular [86], is based on the surfactant capacity associated with their amphipathic structures, which allows some of them to interact and critically destabilize bacterial membranes.

With regard to dipeptides, they are composed of two amino acid residues, but their diversity notably amplifies given the multiple combinatorial options presented by the extended list of available amino acid species, their arrangement in either linear or cyclic structures, and further chemical modifications of their side groups. They are widely spread in nature, having been described within marine ecosystems mostly in microorganisms (Gram-negative and Gram-positive bacteria, cyanobacteria, and fungi) and sponges, but also in eukaryotic algae, invertebrates (e.g., cnidarians, echinoderms, and tunicates), and vertebrates (even mammals) [54,113,114,115]. Among them, the cyclic dipeptides (also known as cyclodipeptides), and their derivatives, belong to the diketopiperazine (DKP) class, particularly to the regioisomeric group of 2,5-DKPs, which result from the connection of their corresponding α-amino acid end groups in a head-to-tail cyclization manner (Figure 2b). This rigid and stable double-lactam ring structure, all formed by peptide bonds, provides them with suitable features for their application in biological systems, as well as with derivatizable side groups for their functional diversification, which altogether may explain their ubiquity in life, especially in marine niches. Despite these facts, not many antimicrobial 2,5-DKPs have been reported from marine organisms so far (Table 1), but they are still considered as a promising source of antimicrobials [113,114,116].

The complexity of natural cyclic peptides exponentially increases with the number of integrating amino acid residues since they allow a greater array of connections, both by overcoming stoichiometric limitations and by providing a greater diversity of potentially combinable functional groups [57,117,118]. A major and widely spread group of bioactive peptides in marine ecosystems is that of depsipeptides. Their members, which exist in both linear and cyclic structures, share the presence of at least one lactone (ester) linkage as a hallmark signature. Bacteria, cyanobacteria, fungi, and sponges appear to be their main producers [55,57,102,116,117,118]. In contrast to 2,5-DKPs, antimicrobial activity (remarkably, often antiviral and mostly tested against HIV1) has been reported for a notable number of depsipeptides, predominantly cyclic depsipeptides (also known as cyclodepsipeptides), some of which occur as the peptidic moiety of lipopeptides such as theopapuamides [110,117]. The molecular mechanisms underlying their antiviral activity are unknown, but several studies have reported them to primarily act by inhibiting the viral entry [101,106]. Some others have associated their activity with the circular structure [99], as well as the content in certain unusual amino acid residues [97,101,102]. Marine antimicrobial cyclodepsipeptides have been mainly identified in bacteria, fungi, bryozoans, tunicates, and, particularly, sponges [55,57,116,117,118]. Callipeltins, celebeside A, koshikamides, homophymine A, mirabamides, neamphamide A, papuamides, stellettapeptins, and theopapuamides are some examples of antimicrobial cyclodepsipeptides found in sponges (see Table 1 for further details) [55,57,102,117].

Among marine antimicrobial cyclodepsipeptides, special attention has been placed on the group of didemnins. Their first members were initially reported as isolated from the Caribbean tunicate (ascidian) *Trididemnum solidum* (family *Didemnidae*) and showed potent antitumoral and antiviral activities, especially didemnin B, which was the first marine natural compound clinically tested in humans (as an anticancer agent just until Phase II due to toxicity issues) [119,120,121]. Further studies demonstrated that these didemnins are actually produced by the symbiotic α-proteobacteria *Tistrella mobilis* and *Tistrella bauzanensis* [121,122]. However, its analog plitidepsin (also known as dehydrodidemnin B, Figure 2d and Table 1), which contains a pyruvic acid instead of the lactic acid in didemnin B, is less toxic and was granted orphan drug status by the FDA in 2004 for the treatment of acute lymphoblastic leukemia and multiple myeloma. Plitidepsin, which is reported to be isolated from the ascidian *Aplidium albicans*, is marketed by PharmaMar S. A. as Aplidin^®^. It acts as an inhibitor of the eukaryotic translation elongation factor 1 alpha 1 (eEF1A) that induces cell cycle arrest leading to growth inhibition and apoptosis. By this mode of action, plitidepsin has also been shown to exert potent antiviral activity against a wide array of viruses, including, very recently, SARS-CoV-2, being under Phase 3 clinical assays currently for the treatment of COVID-19 [107,108].

So far, several factors have limited the discovery of these molecules. For instance, since these compounds are products derived from enzymatic reactions from other substrate molecules, they are not genetically encoded, and, therefore, they have not been able to be identified using the powerful massive sequencing and analysis tools developed in recent years. Fortunately, the latest advances in chromatography, and particularly in their associated detector systems, are rapidly improving the efficiency and speed of molecular profiling. In turn, this is also promoting the metabolomics discipline, which is increasingly contributing to the identification of the organisms’ metabolites (i.e., metabolome), their synthesis pathways, and interactions. Additionally, as mentioned previously, virtual screenings are being implemented to overcome the limitations of high-throughput in vitro testing trials, but also, further initiatives to build libraries of structural formulas of these compounds are necessary to increase their performance. In this sense, it is worth mentioning Bergmann’s pioneering systematic works of profiling marine natural compounds, which were extended by Blunt and Munro to create the MarinLit database [123]. Later, the MarinLit database was acquired and continuously updated by the Royal Society of Chemistry (RSC) (https://marinlit.rsc.org/ (accessed on 27 May 2022)). This library currently contains structural information on 38,689 marine compounds, many of them probably with still undiscovered antimicrobial activity.

## Figures and Tables

**Figure 1 marinedrugs-20-00363-f001:**
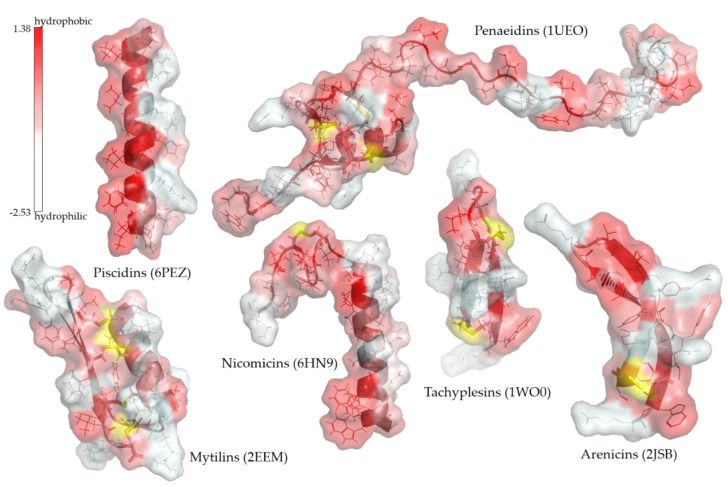
Examples of known molecular structures of some unique marine-AMP class members. The secondary and tertiary structure of various AMPs is presented, including their molecular surface colored according to a gradient of hydrophobicity of their surface amino acids. When present, disulfide bonds are shown in yellow. Amino acid side chains are shown as sticks. Adjacent to each AMP family name is the PDB ID of the structure used. The three-dimensional structure images were generated with PyMOL (The PyMOL Molecular Graphics System, Version 2.3 Schrödinger, LLC, New York, NY, USA).

**Figure 2 marinedrugs-20-00363-f002:**
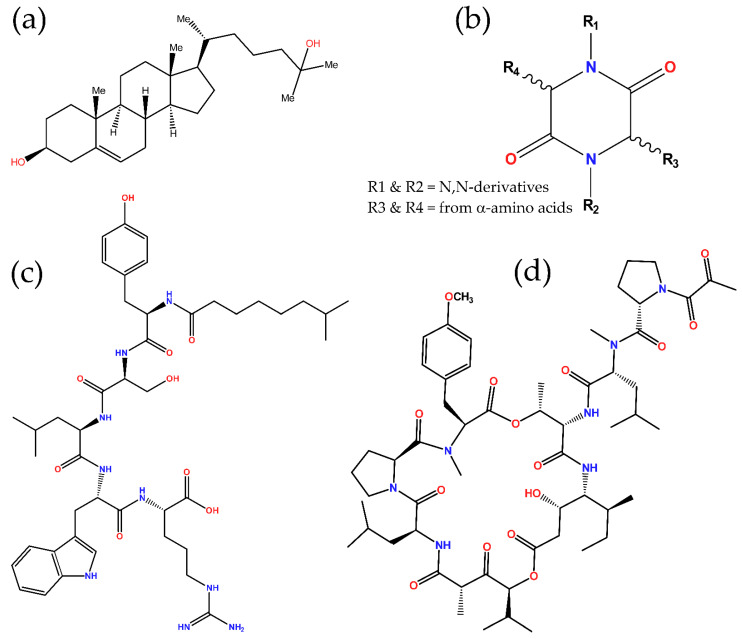
Chemical structures of some antimicrobial metabolites found in marine organisms. Nitrogen and oxygen atoms are colored in blue and red, respectively. (**a**) 25-hydroxycholesterol; (**b**) 2,5-DKP general scaffold; (**c**) tauramamide; and (**d**) plitidepsin.

**Table 1 marinedrugs-20-00363-t001:** Marine antimicrobial metabolites cited in this work classified into their main chemical families.

Antimicrobial Metabolite	Source	Bioactivity	Ref.
**Lipopeptides**			
Aneurinifactin	*Aneurinibacillus aneurinilyticus*	Antibacterial ^a^	[86]
Tauramamide	*Brevibacillus laterosporus*	Antibacterial	[87]
**2,5-DKPs (cyclodipeptides)**			
(3Z,6Z)-3-(4-hydroxybenzylidene)-6- isobutylidenepiperazine-2,5-dione	*Streptomyces* sp.	Antiviral ^b^	[88]
Bacillusamide A	*Bacillus* sp.	Antibacterial, antifungal	[89]
Brevianamide S	*Aspergillus versicolor*	Antibacterial	[90]
Cristatumin A	*Eurotium cristatum*	Antibacterial	[91]
Cyclo(d-6-Hyp-l-Phe), cyclo(l-6-Hyp-l-Phe), and cyclo(6,7-en-Pro-l-Phe)	*Chromocleista* sp.	Antibacterial	[92]
Cyclomarazine A, and cyclomarazine B	*Salinispora arenicola*	Antibacterial ^a^	[93]
Dehydroxybis(dethio)bis(methylthio)gliotoxin	*Pseudallescheria* sp.	Antibacterial	[94]
Etzionin	Unidentified Red Sea tunicate	Antifungal	[95]
**Cyclodepsipeptides (cyclopeptides) from sponges**			
Callipeltin A	*Callipelta* sp.	Antifungal, antiviral	[96]
Celebeside A	*Siliquariaspongia mirabilis*	Antiviral	[97]
Homophymine A	*Homophymia* sp.	Antiviral	[98]
Koshikamides F–H	*Theonella cupola*, and *T. swinhoei*	Antiviral	[99]
Microspinosamide	*Sidonops microspinosa*	Antiviral	[100]
Mirabamides A–D	*S. mirabilis*	Antiviral	[101,102]
Mirabamides E–H	*Stelletta clavosa*	Antiviral	[103]
Neamphamide A	*Neamphius huxleyi*	Antiviral	[104]
Papuamide A–D	*S. mirabilis*, and *T. swinhoei*	Antiviral	[101,105,106]
Plitidepsin (dehydrodidemnin B)	*Aplidium albicans*	Antiviral	[107,108]
Stellettapeptins A–B	*Stelletta* sp.	Antiviral	[109]
Theopapuamides A–D	*S. mirabilis*, and *T. swinhoei*	Antifungal, antiviral	[97,110]

^a^ Reported as active against more than one species. ^b^ Reported as active against influenza A virus (IAV), any other antivirals here were just tested against HIV1.

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
