# Peer review of "The Immune System of Marine Organisms as Source for Drugs against Infectious Diseases"

_marinedrugs, 2022, doi:10.3390/md20060363_

Round 1
Reviewer 1 Report
Review comments to the author
Title: ''The immune system of marine organisms as source for drugs against infectious diseases''.
Manuscript ID: marinedrugs-1744851.
1- Page 7, Figure 2: The resolution and quality of chemical structures should be improved.
Antimicrobial metabolites
1- All antimicrobial metabolites should be listed in a table like this:
Column 1: Secondary metabolite.
Column 2: Source.
Column 3: Bioactivity.
Column 4: Mode of action.
Column 5: References.
2- It is necessary to mention the chemical structure of the antimicrobial metabolites.
Abbreviations:
- List of abbreviations should be inserted by the end of the manuscript before references.
References:
1- The names of all journals should be written uniformly, either abbreviated or written in full form.
2- The first letters of each word in the journal names should be written in uppercase letters (e.g., Ref.2.: Clinical microbiology and infection).
3- All, ''in vivo & in vitro'' words should be typed in italic fonts (e.g., Ref. 57, 60).
4-All scientific names of organisms and species should be typed in italic fonts (e.g., Ref. 58, 60: Oreochromis mossambicus, Propionibacterium acnes, Candida albicans, and Trichomonas vaginalis).
5- Apply all these notes to all references.
Author Response
Point-by-point response to reviewer attached.

Reviewer 2 Report
The Authors in this mini-review have overlooked basic aspects of the biology of antimicrobial peptides(AMPS) in fish and current methodologies that employ biophysical tools with model membranes and bacterial cells to characterize the mode of action of AMPS, and accelerate their therapeutical use.
Almost all organisms produce AMPS.The Authors have not emphasized the importance of AMPS in fish as primary immune molecules, included their families, described the piscidins as potent and broad-spectrum and their conservation among Acanthopterygii superorder, and therapeutical applications among other AMPS.Also they have not provided a tissue distribution of piscidins, their structural properties( a diagrammatic representation for instance) and gene expression that influenced their immune responses, pharmaceutical importance and the biological applications.This review could also explain the broad spectrum of knowledge on piscidins, their classes and types, cytotoxicity, membrane permeabilization(using microscopy to visualize bacteria morphology and peptide location) and properties.
Author Response

(The authors gave the same response as above.)

Round 2
Reviewer 2 Report
The manuscript is now accepted in the present form. I do hope my previous recommendations and comments on the revised manuscript will be potential for the improvement of any advanced future research.